# Conditional Lagrangian Wasserstein Flow for Time Series Imputation

## Abstract

Time series imputation is important for numerous real-world applications. To overcome the limitations of diffusion model-based imputation methods, e.g., slow convergence in inference, we propose a novel method for time series imputation in this work, called Conditional Lagrangian Wasserstein Flow. The proposed method leverages the (conditional) optimal transport theory to learn the probability flow in a simulation-free manner, in which the initial noise, missing data, and observations are treated as the source distribution, target distribution, and conditional information, respectively. According to the principle of least action in Lagrangian mechanics, we learn the velocity by minimizing the corresponding kinetic energy. Moreover, to incorporate more prior information into the model, we parameterize the derivative of a task-specific potential function via a variational autoencoder, and combine it with the base estimator to formulate a Rao-Blackwellized sampler. The propose model allows us to take less intermediate steps to produce high-quality samples for inference compared to existing diffusion methods. Finally, the experimental results on the real-word datasets show that the proposed method achieves competitive performance on time series imputation compared to the state-of-the-art methods.

## 1 Introduction

Time series imputation is essential for various practical scenarios in many fields, such as transportation, environment, and medical care, etc. Deep learning-based approaches, such as RNNs, VAEs, and GANs, have been proved to be advantageous compared to traditional machine learning methods on various complex real-words multivariate time series analysis tasks [18]. More recently, diffusion models, such as denoising diffusion probabilistic models (DDPMs) [20] and score-based generative models (SBGMs) [43], have gained more and more attention in the field of time series analysis due to their powerful modelling capability [26, 32].

Although many diffusion model-based time series imputation approaches have been proposed and show their advantages compared to conventional deep learning models [44, 11, 12], they are limited to slow convergence or large computational costs. Such limitations may prevent them being applied to real-world applications. To address the aforementioned issues, in this work, we leverage the optimal transport theory [47] and Lagrangian mechanics [3] to propose a novel method, called Conditional Lagrangian Wasserstein Flow (CLWF), for fast and accurate time series imputation.

In our method, we treat the multivariate time series imputation task as a conditional optimal transport problem, whereby the random noise is the source distribution, the missing data is the target distribution, and the observed data is the conditional information. To generate new data samples efficiently and accurately, we need to find the shortest path in the probability space according to the optimal transport theory. To this end, we first project the original source and target distributions into the Wasserstein

space via sampling mini-batch OT maps. Afterwards, we construct the time-dependent intermediate samples through interpolating the source distribution and target distribution. Then according to the principle of least action in Lagrangian mechanics [3], the optimal velocity function moving the source distribution to the target distribution is learned in a self-supervised manner by minimizing the corresponding kinetic energy. Moreover, to further improve the model's performance, we learn the task-specific potential function by training a Variational Autoencoder (VAE) model [22] on the observed time series data to build a Rao-Blackwellized trajectory sampler.

Finally, CLWF is assessed on two real-word multivariate time series datasets. The obtained results show that the proposed method achieves competitive performance and admits fast convergence compared with other state-of-the-art time series imputation methods.

The contributions of the paper ares summarized as follows:

- We present Conditional Lagrangian Wasserstein Flow, a novel conditional generative framework based on the optimal transport theory and Lagrangian mechanics;

- We propose a Rao-Blackwellized trajectory sampler to enhance the data generation performance by incorporating the prior information;

- We develop the practical algorithms to solve the time series imputation problem via a conditional generative approach;

- We demonstrate that the proposed method has competitive performance on time series imputation tasks compared other state-of-the-art methods.

# 2 Preliminaries

In this section, we concisely introduce the fundamentals of stochastic differential equations, optimal transport, Shrödinger Bridge, and Lagrangian mechanics.

## 2.1 Stochastic Differential Equations

We treat the data generation task as an initial value problem (IVP), in which $X_0 \in \mathbb{R}^d$ is the initial data (e.g., some random noise) at the initial time $t = 0$, and $X_T \in \mathbb{R}^d$ is target data at the terminal time $t = T$. To solve the IVP, we consider a stochastic differential equation (SDE) defined by a Borel measurable time-dependent drift function $\mu_t : [0, T] \times \mathbb{R}^d \to \mathbb{R}^d$, and a positive Borel measurable time-dependent diffusion function $\sigma_t : [0, T] \to \mathbb{R}^d_{>0}$. Accordingly, the Itô form of the SDE can be described as follows [36]:

$$\mathrm{d}X_t = \mu_t(X_t, t)\mathrm{d}t + \sigma_t \mathrm{d}W_t, \tag{1}$$

where $W_t$ is a Brownian motion/Wiener process. When the diffusion term is not considered, the SDE degenerates to an ordinary differential equation (ODE). However, we will use the SDE for the theoretical analysis as it is more general.

The Fokker–Planck equation (FPE) [40] describing the evolution of the marginal density $p_t(X_t)$ reads:

$$\frac{\partial}{\partial t} p_t(X_t) = -\boldsymbol{\nabla} \cdot (p_t \mu_t) + \frac{\sigma_t^2}{2} \Delta p_t, \tag{2}$$

where $\Delta p_t = \boldsymbol{\nabla} \cdot (\nabla p_t)$ is the Laplacian. In fact, both Eq. eq:sde and Eq. eq:fpe reveal the dynamics of the system and serve as the boundary conditions for the optimization problems we will introduce in later sections with different focuses. The differences are when the constraint is Eq. (1), the formalism is Lagrangian, which depicts the movement of each individual particle; while when the constraint is Eq.(2), the formalism is Eulerian, which depicts the evolution of population.

## 2.2 Optimal Transport

The optimal transport (OT) problem aims to seek the optimal transport plans/ maps that moves the source distribution to the target distribution [47, 41, 38]. In the Kantorovich's formulation of the OT problem, the transport costs are minimized with respect to some probabilistic couplings/joint distributions [47, 41, 38]. Let $p_0$ and $p_T$ be two Borel probability measures with finite second

moments on the space $\Omega \in \mathbb{R}^d$. $\Pi(p_0, p_T)$ denotes a set of transport plans between these two marginals. Then, the Kantorovich's OT problem is defined as follows:

$$\inf_{\pi \in \Pi(p_0, p_T)} \int_{\mathcal{X} \times \mathcal{Y}} \frac{1}{2} \|x - y\|^2 \pi(x, y) \mathrm{d}x \mathrm{d}y, \tag{3}$$

where $\Pi(p_0, p_T) = \{\pi \in \mathcal{P}(\mathcal{X} \times \mathcal{Y}) : (\pi^x)_{\#}\pi = p_0, (\pi^y)_{\#}\pi = p_T\}$, with $\pi^x$ and $\pi^x$ being two projections of $\mathcal{X} \times \mathcal{Y}$ on $\Omega$. The minimizer of Eq .(3), $\pi*$, always exist and referred to as the optimal transport plan.

Note that the R.H.S of Eq. (3) can also include an entropy regularization term $D_{\mathrm{KL}}(\pi \| p_0 \otimes p_T)$, then the original OT problem transforms into the entropy-regularized optimal transport (EROT) problem with Eq. (2) as the constraint, which frames the transport problem better in terms of convexity and stability [13] In particular, from a data generation perspective, $p_0$ is some random initial noise and $p_T$ is the target data distribution, and we can sample the optimal transport maps in a mini-batch manner [46, 45, 39].

## 2.3  Shrödinger Bridge

The transport problem in Sec. 2.2 can be further viewed from a distribution evolution perspective, which is particularly suitable for developing the flow models that model the data generation process. For this reason, the Shrödinger Bridge (SB) problem is introduced [25]. Assume that $\Omega \in C^1([0, T], \mathbb{R}^d)$, $\mathcal{P}(\Omega)$ is a probability path measure on the path space $\Omega$, then the goal of the SB problem aims to find the following optimal path measure:

$$\mathbb{P}* = \arg\min_{\mathbb{P} \in \mathcal{P}(\Omega)} D_{\mathrm{KL}}(\mathbb{P} \| \mathbb{Q}) \quad \text{subject to } \mathbb{P}_0 = q_0 \text{ and } \mathbb{P}_T = q_T, \tag{4}$$

where the Kullback–Leibler (KL) divergence $D_{\mathrm{KL}}(\mathbb{P} \| \mathbb{Q}) = \begin{cases} \log \frac{\mathrm{d}\mathbb{P}}{\mathrm{d}\mathbb{Q}} \mathrm{d}\mathbb{P}, & \text{if } \mathbb{P} \ll \mathbb{Q}, \\ +\infty, & \text{otherwise,} \end{cases}$ and $\mathbb{Q}$ is a reference path measure, e.g., Brownian motion or Ornstein-Uhlenbeck process. Moreover, the distribution matching problem in Eq. (3) can be cast as a dynamical SB problem as well [19, 24, 28]:

$$\arg\min_{\theta} \mathbb{E}_{p(X_t)} \left[ \frac{1}{2} \|\mu_t^\theta(X_t, t)\|^2 \right], \tag{5}$$

$$\text{subject to Eq. (1) or Eq. (2),}$$

where $\theta$ is the parameters of the variational drift function $\mu_t$.

## 2.4  Lagrangian Mechanics

In this section, we formulate the data generation problem under the framework of Lagrangian mechanics [3]. Let $p_t$ and $\dot{p}_t = \frac{\mathrm{d}p_t}{\mathrm{d}t}$ be the density and law of the generalized coordinates $X_t$, respectively. $\mathcal{K}(p_t, \dot{p}_t, t)$ is the kinetic energy, and $\mathcal{U}(p_t, t)$ is the potential energy, then the corresponding Lagrangian is

$$\mathcal{L}(p_t, \dot{p}_t, t) = \mathcal{K}(p_t, \dot{p}_t, t) - \mathcal{U}(p_t). \tag{6}$$

We assume that Eq. (6) is lower semi-continuous (lsc) and strictly convex in $\dot{p}_t$ in the Wasserstein space. The kinetic energy $\mathcal{K}(x_t, \mu_t, t)$ and potential energy $\mathcal{U}(p_t, t)$ are defined as follows, respectively:

$$\mathcal{K}(x_t, \mu_t, t) = \mathbb{E}_{p(X_t)} \left[ \int_0^T \int_{\mathbb{R}_d} \frac{1}{2} \|\mu_t(x_t, t)\|^2 \mathrm{d}x \mathrm{d}t, \tag{7}$$

$$\mathcal{U}(p_t, t) = \mathbb{E}_{p(X_t)} \left[ \int_{\mathbb{R}_d} U_t(X_t) \right] dX_t, \tag{8}$$

where $U_t(X_t)$ is the potential function. Then the *action* in the context of Lagrangian mechanics is defined as follow:

$$\mathcal{A}[\mu_t(x)] = \int_0^T \int_{\mathbb{R}_d} \mathcal{L}(x_t, \mu_t, t) dx_t dt. \tag{9}$$

According to *the principle of least action*, the shortest path is the one minimizing the action, which is aligned with Eq. (4) in the SB theory as well. Therefore, we can leverage the Lagrangian dynamics to tackle the OT problem for data generation. To solve Eq. (6), we need to satisfy the stationary condition, i.e., the Euler-Lagrangian equation:

$$\frac{d}{dt}\frac{\partial}{\partial \dot{p}_t}\mathcal{L}(x_t, \mu_t, t) = \frac{\partial}{\partial p_t}\mathcal{L}(p_t, \dot{p}_t, t), \tag{10}$$

with the boundary condition $\frac{\mathrm{d}X_t}{\mathrm{d}t} = \mu(X_t, t), \; q_0 = p_0, \; q_T = p_T$.

## 3  Conditional Lagrangian Wasserstein Flow for Time Series Imputation

In the section, building upon the optimal transport theory, the Shrödinger Bridge problem, and Lagrangian mechanics introduced in Sec. 2, we propose Conditional Lagrangian Wasserstein Flow, which is a novel conditional generative method for time series imputation.

### 3.1  Time Series Imputation

Our goal is to impute the missing time series data points based on the observations. For training, we adopt adopt a conditionally generative approach for time series imputation in the sample space $\mathbb{R}^{K \times L}$, where $K$ represents the dimension of the multivariate time series and $L$ represents sequence length. In our self-supervised learning approach, the total observed data $x^{\mathrm{obs}} \in \mathbb{R}^{K \times L}$ are partitioned into the imputation target $x^{\mathrm{tar}} \in \mathbb{R}^{K \times L}$ and the conditional data $x^{\mathrm{cond}} \in \mathbb{R}^{K \times L}$.

As a result, the missing data points $x^{\mathrm{tar}}$ can be generated based on the conditions $x^{\mathrm{cond}}$ joint with some uninformative initial distribution $x_0 \in \mathbb{R}^{K \times L}$ (e.g., Gaussian noise) at time $t = 0$, then the imputation task can be described as: $x^{\mathrm{tar}} \sim p(x^{\mathrm{tar}}|x_0^{\mathrm{cond}})$, where the total input of the model is $x_0^{\mathrm{input}} := (x^{\mathrm{cond}}, x_0) \in \mathbb{R}^{K \times L \times 2}$.

### 3.2  Interpolation in Wasserstein Space

To solve Eq. (7), we need to sample the intermediate variable $X_t$ in the Wasserstein space first. To do so, the interpolation method is adopted to construct the intermediate samples. According to the OT and SB problems introduced in Sec. 2, we define the following time-differentiable interpolant:

$$I_t : \Omega \times \Omega \to \Omega \quad \text{such that } I_0 = X_0 \text{ and } I_T = X_T, \tag{11}$$

where $\Omega \in \mathbb{R}^d$ is the support of the marginals $p_0(X_0)$ and $p_T(X_T)$, as well as the conditional $p(X_t|X_0, X_T, t)$.

For implement $I_t$, first, we independently sample some random noise $X_0 \sim \mathcal{N}(0, \sigma_0^2)$ at the initial time $t = 0$ and the data samples $X_T \sim p(x^{\mathrm{tar}})$ at the terminal time $t = T$, respectively. Afterwards, the interpolation method is used to construct the intermediate samples $X_t \sim p(X_t|X_0, X_T, t)$, where $t \sim \mathrm{uniform}(0, T)$ [30, 2, 45]. More specifically, we design the following sampling approach:

$$X_t = \frac{t}{T}(X_T + \gamma_t) + (1 - \frac{t}{T})X_0 + \alpha(t)\sqrt{\frac{t(T-t)}{T}}\epsilon, \quad t \in [0, T], \tag{12}$$

where $\gamma_t \sim \mathcal{N}(0, \sigma_\gamma^2)$ is some random noise with variance $\sigma_\gamma$ injected to the target data samples for improving the coupling's generalization property, and $\alpha(t) \geq 0$ is a time-dependent scalar.

Note that Eq. (12) can only allow us to generate time-dependent intermediate samples in the Euclidean space but not the Wasserstein space, which can lead to slow convergence as the sampling paths are not straightened. Hence, to address this issue, we need to project the interpolations in the Wasserstein space before interpolating to strengthen the probability flow. To this end, we leverage the method adopted in [46, 45, 39] to sample the optimal mini-batch OT maps between $X_0$ and $X_T$ first, and perform the interpolations according to Eq. (12) afterwards. Finally, we have the joint variable $x_t^{\mathrm{input}} := (x^{\mathrm{cond}}, x_t)$ as the input for computing the velocity of the Wasserstein flow.

### 3.3  Flow Matching

To estimate the velocity of the Wasserstein flow $\mu_t(X_t, t)$ in Eq. (1), the previous methods that require trajectory simulation for training can result in long convergence time and large computational costs

[9, 37]. To circumvent the above issues, in this work we adopt a simulation-free training strategy based on the OT theory introduce in Sec. 2.2 [30, 46, 2], which turns out to be faster and more scalable to large time series datasets.

Since we can now draw mini-batch interpolated samples of the source distribution and target distribution in the Wasserstein space using Eq. (12), we can model the variational velocity function using a neural network with parameters $\theta$. Then, according to Eq. (1), the target velocity can be computed by the difference between the source distribution and target distribution. Therefore, the variational velocity function $\mu_\theta(x_t^{\text{input}}, t)$ can be learned by

$$\arg\min_\theta \int_0^T \int_{\mathbb{R}^d \times \mathbb{R}^d \times \mathbb{R}^d} \left\| \frac{\mathrm{d}X_t}{\mathrm{d}t} - \mu_t^\theta(x_t^{\text{input}}, t) \right\|^2 \mathrm{d}x_0 \mathrm{d}x^{\text{tar}} \mathrm{d}x^{\text{input}} \mathrm{d}t \tag{13}$$

$$\approx \arg\min_\theta \mathbb{E}_{p(x_0), p(x^{\text{tar}}), p(x^{\text{input}}), t} \left[ \left\| \frac{x^{\text{tar}} - x_0}{T} - \mu_t^\theta(x_t^{\text{input}}, t) \right\|^2 \right]. \tag{14}$$

Eq. (14) can be solved by drawing mini-batch samples in the Wasserstein space and performing stochastic gradient descent accordingly. In this fashion, the learning process is simulation-free as the trajectory simulation is not needed.

Moreover, note that that Eq. (13) also obeys the principle of least action introduced in Sec. 2.4 as it minimizes the kinetic energy described in Eq. (7). Therefore, it indicates that the geodesic that drives the particles from the source distribution to the target distribution in the OT problem described in Sec. 2 is found as well, which enables us to generate new samples with less simulation steps compared to standard diffusion models.

### 3.4 Potential Function

So far, we have demonstrated how to leverage the kinetic energy to estimate the velocity in the Lagrangian described by Eq. 6. Apart from this, we can also incorporate the prior knowledge within the task-specific potential energy into the dynamics, which enables us to further improve the data generation performance. To this end, let $U(X_t) : \mathbb{R}^d \times [0, T] \to \mathbb{R}$ be the task-specific potential function depending on the generalized coordinates $X_t$ [48, 37, 34]. Therefore, we can compute the dynamics of the system by

$$\frac{\mathrm{d}X_t}{\mathrm{d}t} = v_t(X_t, t) = -\nabla_x U_t(X_t). \tag{15}$$

Since the data generation problem in our case can also be interpreted as a stochastic optimal control (SOC) problem [4, 17, 35, 50, 21, 5], then the existence of such $U_t(X_t)$ is assured by Pontryagin's Maximum Principle (PMP) [16].

To estimate $v_t(X_t, t)$, according to the Lagrangian Eq. (6), we assume that the potential function takes the form $U_t(X_t) \approx -\log \mathcal{N}(X_t | \hat{X}_t, \sigma_p^2)$, where $\hat{X}_t$ the learned mean and $\sigma_p^2$ is the pre-defined variance. As a result, the derivative is $\nabla_x U(X_t) = \frac{X_t - \hat{X}_t}{\sigma_p^2}$. In terms of practical implementation, we parameterize $\nabla_x U(X_t)$ via a Variational Autoencoder (VAE) [22]. More specifically, we pre-train a VAE on the total observed time series data $X^{\text{obs}}$. Afterwards, the reconstruction discrepancies of the VAE are used to approximate the task-specific $v^\phi(X_t, t)$ depending on $X_t$:

$$v_t^\phi(X_t, t) = -\frac{1}{\sigma_p^2}(X_t - \text{VAE}(X_t)), \tag{16}$$

where $\text{VAE}(X_t)$ represents the reconstruction output of the pre-trained VAE model with input $X_t$, and $\sigma_p^2$ is treated as a positive constant for simplicity. In this manner, we can incorporate the prior knowledge learned from the accessible training data into the sampling process formulated by Eq. (14) to enhance the data generation performance.

### 3.5 Rao-Blackwellized Sampler

To generate the missing time series datapoints, we first formulate an unbiased ODE sampler $S(X_t, \mu_t^\theta(X_t, t), t)$ for $X_{t+1}$ with the Euler method and $\mu_t^\theta(X_t, t)$ learned by Eq. (14) (which

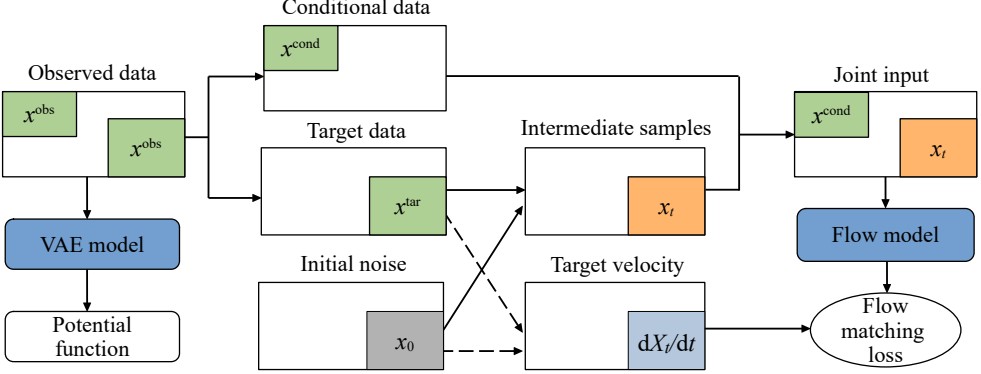

Figure 1: The overall training process of Conditional Lagrangian Wasserstein Flow.

means the diffusion term in Eq. 1 is omitted). Alternatively, one can also adopt the SDE sampler by using the Euler–Maruyama method. Nevertheless, to ensure achieve the best imputation performance, we choose the ODE sampler for implementation. Note that the ODE sampler alone is good enough to generate high-quality samples for time series imputation.

Now we can construct a Rao-Blackwellized trajectory sampler [8] for time series data imputation using Eq. 14 and Eq. 16. To this end, we first treat $\mathcal{S}(X_{t+1}|X_t, \mu_t^\theta(X_t, t), t)$ be the base estimator for $X_{t+1}$ with $\mathbb{E}[\mathcal{S}^2] < \infty$ for all $X_{t+1}$. And we assume $\mathcal{T}(X_t, v_t^\phi(x_t, t), t)$ is a sufficient statistic for $X_{t+1}$ based on Eq. 16, even it is not a very accurate estimator for $X_{t+1}$. As a result, we can formulate a new trajectory sampler $\mathcal{S}^* = \mathbb{E}[\mathcal{S}|\mathcal{T}]$ to generate the missing time series data. Then according to the Rao-Blackwell theorem [8], we have

$$\mathbb{E}[\mathcal{S}^* - X_{t+1}]^2 \leq \mathbb{E}[\mathcal{S} - X_{t+1}]^2, \tag{17}$$

where the inequality is strict unless $\mathcal{S}$ is a function of $\mathcal{T}$. Eq. 17 suggests we can construct a more powerful sampler with smaller errors than the base ODE sampler $\mathcal{S}$ using Rao-Blackwellization.

## 3.6 The Algorithms

The overall training process of CLWF is illustrated in Fig. 1, which consists of the following stages. First, the total observed data $x^{\text{obs}}$ are partitioned into the target data and conditional data for training. Next, the data pairs of $x^{\text{tar}}$ and $x_0$ are sampled from the target dataset and random Gaussian noise, respectively. Then, the data pairs are projected into the Wasserstein space by sampling the corresponding OT maps. After that, the intermediate variable $x_t$ is sampled through interpolation using Eq. (12). We can approximate the target velocity $\frac{\mathrm{d}X_t}{\mathrm{d}t}$ by computing $\frac{x^{\text{tar}} - x_0}{T}$. Subsequently, we use the joint distribution of the conditional information $x^{\text{cond}}$ and the intermediate variable $x_t$, $x^{\text{input}}$ as the total input to feed the variational flow model $\mu_t^\theta$ to compute the velocity. And the flow matching loss defined by Eq. (14) is minimized by stochastic gradient descent.

Furthermore, to incorporate the prior information of into the model, we can choose to train a VAE model on the total observed data $x^{\text{obs}}$. This is used to estimate the derivative of the task-specific potential function according to Eq. (16), which can be further utilized to construct a more powerful Rao-Blackwellized sampler for inference.

For inference, at time $t = 0$, we sample the initial random noise $x_0$ and conditional information $x^{\text{cond}}$ to formulate the joint variable $x^{\text{input}}$. Note that during the trajectory sampling $x_0$ will evolve over time, while $x^{\text{cond}}$ remain invariant. We use $x^{\text{input}}$ as the input of the flow model $\mu_t^\theta$ to compute the velocity. Afterwards, we sample the new $x_t$ using the Euler method. If we perform Rao-Blackwellization, then $x_t$ is fed to the VAE model for computing the derivative of the potential function, and $x_t$ is updated again using the Euler method. The above process will be repeated until reach its convergence. Moreover, we can sample multiple trajectories using different initial random

noise, and the averages as the final imputation results. Finally, the proposed training and sampling procedures are presented in Algorithm 1 and Algorithm 2, respectively.

---

**Algorithm 1** Training procedure

---

**Require:** Terminal time: $T$, max epochs, observed data $X^{\text{obs}}$, parameters: $\theta$ and $\phi$.
    **while** epoch < max epochs **do**
        sample $t$, $(x_0, x_T)$, and OT maps;
        sample $x_t$ according to Eq. (12);
        minimize Eq. (14);
    **end while**
    **if** Rao-Blackwellization **then**
        train a VAE model on $X^{\text{obs}}$.
    **end if**

---

**Algorithm 2** Sampling procedure

---

**Require:** initial time $t = 0$, terminal time: $T = 1$, Euler method step number: $N$.
    sample initial noise $X_0 \sim \mathcal{N}(0, \sigma_0^2)$
    **while** $t < $ T **do**
        $X_t = X_t + \mu_t^\theta(x^{\text{input}}, t)\frac{T}{N}$
        **if** Rao-Blackwellization **then**
            $X_t = X_t + v_t^\phi(x_t, t)\frac{T}{N}$
        **end if**
        $t = t + \frac{T}{N}$
    **end while**

---

# 4 Experiments

## 4.1 Datasets

We use two public multivariate time series datasets for validation. The first dataset is the PM 2.5 dataset [51] from the air quality monitoring sites for 12 months. The missing rate of the raw data is 13%. The feature number $K$ is 36 and the sequence length $L$ is 36. In our experiments, only the observed datapoints are masked randomly as the imputation targets.

The other dataset we use is the PhysioNet dataset [42] collected from the intensive care unit for 48 hours. The feature number $K$ is 35 and the sequence length $L$ is 48. The missing rate of the raw data is 80%. In our experiments, 10% and 50% of the datapoints are masked randomly as the imputation targets, which are denoted as PhysioNet 0.1 and PhysioNet 0.5, respectively.

## 4.2 Baselines

For comparison, we select the following state-of-the-art timer series imputation methods as the baselines: 1) GP-VAE [18], which is combines a VAE model and a Gaussian Process prior; 2) CSDI [44], which is based on the conditional diffusion model; 3) CSBI [12], which is based on the Schrödinger bridge diffusion model; 4) DSPD-GP [7], which combines the diffusion model with the Gaussian Process prior.

## 4.3 Experimental Settings

In terms of the choices of architectures, tboth the flow model and the VAE model are built upon Transformers [44]. We use the ODE sampler for inference and sample the exact optimal transport maps for interpolations to achieve the optimal performance. The optimizer is Adam and the learning rate: 0.001 with linear scheduler. The maximum training epochs is 200. The mini batch size for training is 64. The total step number of the Euler method used in CLWF is 15, while the total step numbers for other diffusion models. i.e., is CSDI, CSBI, and DSPD-GP are 15 (as suggested in their papers). The number of the Monte Carlo samples for inference is 50. The standard deviation $\sigma_0$ for the initial noise $X_0$ is 0.1, and the standard deviation $\sigma_\gamma$ for the injected noise $\gamma_t$ 0.001. The coefficient $\sigma_p^2$ in the derivative of the potential function is 0.01.

## 4.4 Experimental Results

### 4.4.1 Imputation Results

We assess the proposed method on PM 2.5, PhysioNet 0.1 and PhysioNet 0.5, respectively. The root means squared error (RMSE) and mean absolute error (MAE) are used as the evaluation metrics. From the test results shown in Table 1 and Fig. 2, we can see that our method CLWF outperforms the existing deep learning-based method (GP-VAE) and the recent state-of-the-art diffusion methods (CSDI, CSBI, and DSPD-GP). Moreover, CLWF uses only 15 sampling steps for inference, while the

Table 1: Test imputation results on PM 2.5, PhysioNet 0.1, and PhysioNet 0.5 (5-trial averages). The best are in bold and the second best are underlined.

| Method | PM 2.5 | | PhysioNet 0.1 | | PhysioNet 0.5 | |
|---|---|---|---|---|---|---|
| | RMSE | MAE | RMSE | MAE | RMSE | MAE |
| GP-VAE | 43.1 | 26.4 | 0.73 | 0.42 | 0.76 | 0.47 |
| CSDI | 19.3 | 9.86 | 0.57 | 0.24 | 0.65 | 0.32 |
| CSBI | 19.0 | 9.80 | 0.55 | 0.23 | **0.63** | 0.31 |
| DSPD-GP | 18.3 | **9.70** | 0.54 | **0.22** | 0.68 | 0.30 |
| CLWF | **18.1** | **9.70** | **0.47** | **0.22** | 0.64 | **0.29** |

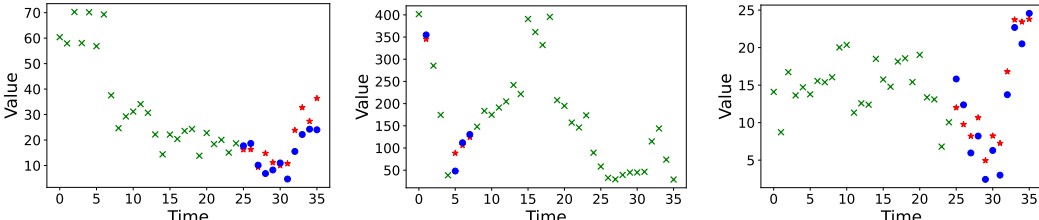

Figure 2: Visualization of the test imputation results on PM 2.5, green dots are the conditions, blue dots are the imputation results, and red dots are the ground truth.

Table 2: Single-sample test imputation results on PM 2.5, PhysioNet 0.1, and PhysioNet 0.5 (5-trial averages).

| Method | PM 2.5 | | PhysioNet 0.1 | | PhysioNet 0.5 | |
|---|---|---|---|---|---|---|
| | RMSE | MAE | RMSE | MAE | RMSE | MAE |
| CSDI | 22.2 | 11.7 | 0.74 | 0.30 | 0.83 | 0.40 |
| CLWF | 18.4 | 10.0 | 0.48 | 0.22 | 0.64 | 0.30 |

baseline diffusion method uses only 50 sampling steps. This suggests that CLWF is faster and more accurate than the existing methods on time series imputation tasks.

### 4.4.2 Ablation Study

**Single-sample Imputation Result.** We compare the time series imputation performance of CLWF with CSDI using only one Monte Carlo sample. The test results shown in Table 2 shows that CWFL outperforms CSDI, which suggests that CWFL exhibits lower imputation variances compared to diffusion-based models. This indicates that CWFL is more efficient and computationally economical for inference.

**Effect of Rao-Blackwellzation.** We compare the test imputation CLWF wth and without using Rao-Blackwellzation. Note that the PhysioNet dataset does not have enough non-zero data points to train a valid VAE model, therefore we only construct the Rao-Blackwellized sampler for the PM 2.5 dataset. The results showed in Table 3 indicates

Table 3: Test imputation results on PM 2.5 (5-trial averages).

| Method | PM 2.5 | |
|---|---|---|
| | RMSE | MAE |
| CLWF (without RB) | 18.2 | 9.75 |
| CLWF (RB) | 18.1 | 9.70 |

that the Rao-Blackwellized sampler can further improve the time series imputation performance of the base sampler.

# 5 Related Work

## 5.1 Diffusion Models

Diffusion models, such as DDPMs [20] and SBGM [43], are considered as the new contenders to GANs on data generation tasks. But they generally take relatively long time to produce high quality samples. To mitigate this problem, the flowing matching methods have been proposed from an optimal transport. For example, ENOT uses the saddle point reformulation of the OT problem to develop a new diffusion model [19] The flowing matching methods have also been proposed based on the OT theory [27, 29, 31, 2, 1]. In particular, mini-batch couplings are proposed to straighten the probability flows for fast inference [39, 45, 46].

The Schrödinger Bridge have also been applied to diffusion models for improving the data generation performance of diffusion models. Diffusion Schrödinger Bridge utilizes the Iterative Proportional Fitting (IPF) method to solve the SB problem [14]. SB-FBSDE proposes to use forward-backward (FB) SDE theory to solve the SB problem through likelihood training [10]. GSBM formulates a generalized Schrödinger Bridge matching framework by including the task-specific state costs for various data generation tasks [28] NLSB chooses to model the potential function rather than the velocity function to solve the Lagrangian SB problem [24]. Action Matching [33, 34] leverages the principle of least action in Lagrangian mechanics to implicitly model the velocity function for trajectory inference. Another classes of diffusion models have also been proposed from an stochastic optimal control perspective by solving the HJB-PDEs [35, 50, 5, 28].

## 5.2 Time Series Imputation

Many diffusion-based models have been recently proposed for time series imputation [26, 32]. For instance, CSDI [44] combines a conditional DDPM with a Transformer model to impute time series data. CSBI [12] adopts the FB-SDE theory to train the conditional Schrödinger bridge model to for probabilistic time series imputation. To model the dynamics of time series from irregular sampled data, DSPD-GP [7] uses a Gaussian process as the noise generator. TDdiff [23] utilizes self guidance and learned implicit probability density to improve the time series imputation performance of the diffusion models. However, the time series imputation methods mentioned above exhibit common issues, such as slow convergence, similar to many diffusion models. Therefore, in this work, we proposed CLWF to tackle thess challenges.

# 6 Conclusion, Limitation, and Broader Impact

In this work, we proposed CLWF, a novel time series imputation method based on the optimal transport theory and Lagrangian mechanics. To generate the missing time series data, following the principle of least action, CLWF learns a velocity field by minimizing the kinetic energy to move the initial random noise to the target distribution. Moreover, we can also estimate the derivative of a potential function via a VAE model trained on the observed training data to further improve the performance of the base sampler by Rao-Blackwellization. In contrast with previous diffusion-based models, the proposed requires less simulation steps and Monet Carlo samples to produce high-quality data, which leads to fast inference. For validation, CWLF is assessed on two public datasets and achieves competitive results compared with existing methods.

One limitation of CLWF is that the samples obtained are not diverse enough as we use ODE for inference, which results in slightly higher test (continuous ranked probability score) CRPS compared to previous works, e.g., CSDI. Therefore, for future work, we will seek suitable approaches to accurately model the diffusion term in the SDE. Moreover, we will also try to design better task-specific potential functions for sparse multivariate time series data. We plan to explore the potential of the Lagrangian Wasserstein Flow model for other time series analysis tasks, such as anomaly detection and uncertainty quantification.

In terms of broader impact, our study on time series imputation has the potential to address important real-world challenges and consequently make a positive impact on daily lives.

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

## A  Stochastic Optimal Control

The data generation task can also be interpreted as a stochastic optimal control (SOC) problem [4, 17, 35, 50, 21, 5] whose cost function $\mathcal{J}$ is defined as:

$$\mathcal{J}(X_t, t) = \mathbb{E}_{p(X_t)}\left[\int_0^T \int_{\mathbb{R}_d} \frac{1}{2}\|\nabla_x \Psi(X_t, t)\|^2 \mathrm{d}X_t \mathrm{d}t\right] + \mathbb{E}_{p(X_T)}\big[\Psi(X_T)\big], \tag{18}$$

where $\frac{1}{2}\|\nabla_x \Psi(X_t, t)\|^2$ denotes the running cost, and $\Psi(X_T)$ denotes the terminal cost. The above SOC problem can be solved by dynamic programming [4, 6].

Further, let $V(X_t, t) = \inf \mathcal{J}(X_t, t)$ be the value function/optimal-cost-to-go of the SOC problem, then the corresponding Hamilton-Jacobi-Bellman (HJB) partial differential equation (PDE) [15, 49] is given by

$$\frac{\partial V_t}{\partial t} - \frac{1}{2}\nabla V_t' \nabla V_t + \frac{1}{2}\Delta V_t = 0, \tag{19}$$

with the terminal condition: $V(X_t, T) = \Psi(X_t)$. $\tag{20}$

## B  Rao-Blackwell Theorem

**Theorem 1** *(Rao-Blackwell) Let $\mathcal{S}$ be an unbiased estimator of some parameter $\theta \in \Theta$, and $\mathcal{T}(X)$ the sufficient statistic for $\theta$, then: 1) $\mathcal{S}^* = \mathbb{E}[\mathcal{S}|\mathcal{T}(X)]$, is an unbiased estimator for $\theta$, and 2) $\mathbb{V}_\theta[\mathcal{S}^*] \leq \mathbb{V}_\theta[\mathcal{S}]$ for all $\theta$. The inequality is strict unless $\mathcal{S}$ is a function of $\mathcal{T}$.*

## C  Experimental Environment

For the hardware environment of the experiments, we use a single NVIDIA A100-PCIE-40GB GPU and an Intel(R) Xeon(R) Gold-6248R-3.00GHz CPU. For the software environment, the Python version is 3.9.7, the CUDA version 11.7, and the Pytorch version is 2.0.1.

