# OpenReview forum: "Conditional Lagrangian Wasserstein Flow for Time Series Imputation"
_NeurIPS.cc/2024/Conference — Submitted to NeurIPS 2024_

### Official Review · Reviewer_MvNs · 2024-07-09

**Soundness:** 3
**Presentation:** 3
**Contribution:** 3
**Rating:** 6
**Confidence:** 4

**Summary:**

The authors introduce the conditional Lagrangian Wasserstein flow method for time series imputation.
The time series imputation task is treated as a conditional data generation problem. The authors use flow matching to learn an ODE sampler for generating the missing time series data. They further propose to enhance the imputation performance via Rao-Blackwellization. The method is tested on the real-world datasets.

**Strengths:**

S1. The approach of using Lagrangian Wasserstein flow for time series imputation is interesting and novel.

S2. Compared to other diffusion model-based time series imputation methods, this method seems more efficient.

S3. The experimental results given in the paper are satisfactory.

**Weaknesses:**

W1. The imputation process is implemented using an ODE sampler. Why not use an SDE sampler?

W2. Why use the Euler method instead of other higher-order solvers, such as the Runge-Kutta methods, for solving the ODE?

W3. Can the proposed method also be applied to time series forecasting tasks?

W4. The dynamic described in Eq (15) is deterministic, which is inconsistent with Eq (1). Why the diffusion term is missing in Eq (15)?

**Questions:**

Please see my weaknesses.

And there is another question,
Q. Can the VAE-based potential function be used alone for time series imputation?

**Limitations:**

CRPS is not used as the evaluation metric in the experiments.

---

> ### Author Rebuttal · Authors · 2024-08-04
>
> Many thanks for the insighful comments.
>
> ## Weakness 1:
> Our proposed method can also be implemented via an SDE sampler by adding a time-dependent volatility term (please see Eq. (12), Sec 3.2, Page 4), but in order to achieve the optimal empirical performance in terms of RMSE and MAE on the time series imputation tasks, we opt for an ODE sampler.
>
>
>
> ## Weakness 2:
> In practice, we found that the Euler method has similar performance to other numerical methods while requiring less memory and computational time. Therefore, we chose to use the Euler method in our implementation.
>
>
>
> ## Weakness 3:
> Yes, the proposed method can be used for time series forecasting tasks as well.
> To this end, we can view the forecasting task as a conditional generation problem, in which one can just use the future states as the data generation target and the current state as the conditional information.
>
>
>
> ## Weakness 4:
> Eq. (15) is derived based on the assumption of an overdamped Langevin dynamic system, which still represents the drift term in Eq.(1).
> For computational convenience, we choose to omit the diffusion term to formulate an ODE sampler. Nevertheless, the theoretical analysis given in the paper still applies to this case.
>
>
>
> ## Questions:
> Although the VAE-based potential function can be seen as a sufficient statistic for the imputation target, it is generally not accurate enough to be used alone to represent the data generation dynamics in time series imputation tasks.
>
>
>
> ## Limitations:
> Since our method is implemented via an ODE sampler CRPS may not be a suitable evaluation metric.
> Furthermore, [1] and [2] point out that CRPS may overlook the model's performance on each dimension, which could lead to an incorrect assessment of overall model performance.
>
>
> Reference:
> [1] Koochali, Alireza, Peter Schichtel, Andreas Dengel, and Sheraz Ahmed. "Random noise vs. state-of-the-art probabilistic forecasting methods: A case study on CRPS-Sum discrimination ability." Applied Sciences 12, no. 10 (2022): 5104.
>
>
> [2] Biloš, Marin, Kashif Rasul, Anderson Schneider, Yuriy Nevmyvaka, and Stephan Günnemann. "Modeling temporal data as continuous functions with stochastic process diffusion." In International Conference on Machine Learning, pp. 2452-2470. PMLR, 2023.

---

> > ### Comment · Reviewer_MvNs · 2024-08-14
> > **Thanks for rebuttal**
> >
> > Thanks for your rebuttal. The authors have addressed all my concerns and I will maintain my score at this stage.

---

### Official Review · Reviewer_FYKM · 2024-07-10

**Soundness:** 3
**Presentation:** 3
**Contribution:** 3
**Rating:** 7
**Confidence:** 4

**Summary:**

In this paper, the authors introduce a new time series imputation model based on the conditional Lagrangian Wasserstein flow. Different from previous diffusion-based models, the proposed model leverages the optimal transport theory and Lagrangian dynamics to improve the data generation performance.

**Strengths:**

1. The overall presentation of paper is clear.
2. The idea of using the conditional Lagrangian Wasserstein flow for time series imputation is novel.
3. The proposed method is assessed on the real-world datasets and shows competitive performance compared to the state-of-the-art methods.

**Weaknesses:**

1. In Sec 3.2, the technique used for projecting the interpolants into the Wasserstein space is unclear. The authors should elaborate more on this.
2. Compared to CSDI and other diffusion-based imputation methods, the proposed method’s data generation process seems deterministic. Hence, it cannot be used to quantify the uncertainty of the prediction.
3. The proposed sampler is implemented through an ODE, which is inconsistent with the Schrodinger Bridge problem in sec 2.3.
4. There are some inconsistencies in the notations, please double-check.

**Questions:**

What is the difference between the data generation processes in Euclidean space and Wasserstein space?

**Limitations:**

The ablation study on the effectiveness of Rao-Blackwellization should be conducted on more than one dataset (PM 2.5).

---

> ### Author Rebuttal · Authors · 2024-08-04
>
> Many thanks for the valuable feedback .
>
> ## Weakness 1:
> To project the intermediate samples from the Euclidean space to the Wasserstein space, We adopt the technique descried in [1] by drawing mini-batches of the initial noise and target samples to compute the corresponding optimal transport maps.
> In terms of Wasserstein metrics, we use the earth mover distance (EMD) for compute the deterministic optimal transport maps, which turns out to perform better than other metrics.
> Alternatively, we can also adopt the Sinkhorn distance to compute the stochastic optimal transport maps with entropy regularization.
> For the practical implementation, we use the Python Optimal Transport (POT) library [2].
>
>
>
> ## Weakness 2:
> This is because we adopt an ODE sampler in our method.
> Although ODE samplers are deterministic, they have other advantages compared to SDE samplers.
> For example, they are usually easier to implement than SDE samplers.
> Moreover, compared to CSDI and other diffusion-based methods, our method requires less simulation steps and samples to achieve good performance on time series imputation tasks.
>
>
>
> ## Weakness 3:
> We can also implement the proposed framework via an SDE sampler.
> The optimization objective becomes the KL divergence between the target SDE and the prior Wiener process, which is aligned with the frame work of the Schrodinger Bridge problem.
> However, in practice, we found that the ODE sampler has better empirical performance in terms of RMSE and MAE compared to the SDE sampler.
>
>
>
> ## Weakness 4:
> We have proofread the manuscript again and corrected the typos.
>
>
>
> ## Questions:
> Since the Euclidean distance is not suitable for measuring the distance between two probabilistic distributions, meaning that we cannot find the shortest path  between  =the initial distribution and the target distribution.
> As a result, performing the data generation processes in Euclidean space may lead to slow convergence and poor sample quality.
> Therefore, we need to perform the data generation processes in the Euclidean space instead of the Wasserstein space.
>
>
>
> ## Limitations:
> We conduct additional experiment using the synthetic dataset.
> And the experimental results demonstrates the effectiveness of the Rao-Blackwellization.  Please take a look at the uploaded PDF for the details.
>
>
> Test imputation results on synthetic data ($5$ trials, values are multiplied by $10^2$.
>
>
> | Method  ||   Synthetic 0.4   ||  Synthetic 0.6  ||  Synthetic 0.8    |
> | ----------- |----------- | ----------- | ----------- | -----------   | -----------  | -----------  |
> |   | RMSE |  MAE  |RMSE |  MAE  |RMSE |  MAE  |
> | CLWF(no RB) |$22.91\pm0.49$ |$15.28\pm0.21$|$25.65\pm0.31$ |$15.54\pm0.22$|$27.41\pm0.27$  |$15.91\pm0.23$ |
> |CLWF(with RB)  | $22.72\pm0.48$| $13.23\pm0.42$|$25.44\pm0.30$|$15.28\pm0.17$| $27.32\pm0.27$| $15.79\pm0.23$ |
>
>
>
> ## References:
>
> [1] Alexander Tong, Kilian FATRAS, Nikolay Malkin, Guillaume Huguet, Yanlei Zhang, Jarrid Rector-Brooks, Guy Wolf, and Yoshua Bengio. Improving and generalizing flow-based genera431 tive models with minibatch optimal transport. Transactions on Machine Learning Research,432, 2024.
>
>
> [2] RFlamary, Rémi, Nicolas Courty, Alexandre Gramfort, Mokhtar Z. Alaya, Aurélie Boisbunon, Stanislas Chambon, Laetitia Chapel et al. "Pot: Python optimal transport." Journal of Machine Learning Research 22, no. 78 (2021): 1-8. Website: https://pythonot.github.io/

---

> > ### Comment · Reviewer_FYKM · 2024-08-12
> >
> > Since all of my questions have been adequately answered by the authors, I recommend acceptance.

---

### Official Review · Reviewer_4rvE · 2024-07-11

**Soundness:** 3
**Presentation:** 3
**Contribution:** 3
**Rating:** 6
**Confidence:** 5

**Summary:**

Inspired by optimal transport and Lagrangian dynamics, this work proposes to use the conditional Lagrangian Wasserstein flow to impute time series data. The method requires less model evaluation steps to generate high quality samples compared to existing diffusion models. Moreover, a task-specific energy function is used to further improve the model’s performance.

**Strengths:**

- The contributions of this paper are significant.
- This work is built on a solid theoretical foundation.
- The experimental results and relevant ablation study findings are provided.

**Weaknesses:**

- How should the values of the hyperparameters (e.g., the variance of the potential function) be chosen? How will these choice affect the model’s performance?
- The reason for choosing the VAE model to construct the potential function is unclear. Are there other functions can also be used as the task-specific potential function?
- The authors proposed to use a VAE model to enhance the data sampling procedure. However, if the VAE model performs poorly, can it still help with the data generation process?

**Questions:**

Can the proposed method be used to compute the likelihood of the generated data samples?

**Limitations:**

The deterministic sampling process may introduce bias, potentially failing to accurately reflect the target marginal distribution.

---

> ### Author Rebuttal · Authors · 2024-08-04
>
> Many thanks for the insightful comments.
>
>
> ## Weakness 1:
> The value of the potential function's variance do effect the Rao-Blackwellized sampler.
> If the variance is too small the effect of the Rao-Blackwellization is negligible, if the variance is too large the performance will decrease.
> In future research, we plan to explore more advanced methods for formulating the potential function for time series imputation tasks.
>
>
> ## Weakness 2:
> Since the VAE model is trained on clean target data, we can leverage its denoising/self-correction effect to force the reconstructed intermediate samples closer to the ground truth during the sampling process.
> An ideal choice of the potential function should be convex.
>
>
>
> ## Weakness 3:
> If the VAE model's performance is too poor, it cannot be considered a sufficient statistic and, consequently, will not aid in the data generation process.
>
>
> ## Questions:
> Unfortunately, in the current implementation we cannot compute the likelihood of the samples obtained because the sampler is deterministic.
> We will leave this problem for future research.
>
>
>
> ## Limitations:
> This is true, although we tried the methods proposed in existing literature, we do not find a satisfactory method to estimate the diffusion term to achieve competitive performance on time series imputation tasks.
> Therefore, we plan to explore better methods for estimating the diffusion term in the flow matching model as part of our future work.

---

### Official Review · Reviewer_3hN3 · 2024-07-12

**Soundness:** 1
**Presentation:** 2
**Contribution:** 1
**Rating:** 2
**Confidence:** 4

**Summary:**

This work trains a conditional flow model from noise to time series data. A VAE is used to estimate the data density then perform interleaved flow and density gradient ascent steps to generate new time series. This is referred to as a Rao-Blackwellization procedure. It is shown empirically that the model performs well on time series imputation tasks across two datasets, and that the VAE-based guidance helps in one case.

**Strengths:**

* Originality: Time series imputation is an important problem and I have not seen any flow matching works in towards this application. Presents a new improvement to the standard flow matching framework with a “Rao-Blackwellization”.

**Weaknesses:**

- Quality: Experiments are limited in scope and the it is not clear whether the “Rao-Blackwellization” (the main methodological novelty) step reliably improves performance.
    - In order to claim “Less sampling steps” it would be great to understand how performance changes with the number of steps for both diffusion and flow-based models.
    - More than 2 time series datasets would allow an understanding of when this “Rao-Blackwellized” sampler is valid.
    - No error bars (even though this is claimed in the checklist)
    - The empirical support is lacking for the effectiveness of the “Rao-Blackwellization” step. Currently it is shown to slightly benefit (although without error bars this is difficult to tell how much) on a single dataset. In this case, an improvement in RMSE from 18.2 to 18.1 on a single dataset seems insignificant without additional detail. While it is claimed that “the PhysioNet dataset does not have enough non-zero datapoints to train a valid VAE model”, a Rao-Blackwellized sampler should always work regardless of the performance of the estimate? It would be good to see if it helps at all on this dataset and others.
- Clarity: It is not clear to me that this even is a “Rao-Blackwellized” sampler. As far as I can tell this “Rao-Blackwellized” sampler is fundamentally biased (while the original sampler is not), and it therefore cannot possibly be a Rao-Blackwellized sampler. It would be great if the authors could prove that Algorithm 2 is really a Rao-Blackwellization (at least under some conditions).
- Significance: Without further experimental benchmarking especially with regards to the novel sampler, the applicability of this method is very limited.

**Questions:**

How is this a Rao-Blackwellization, and what does the paper

Minor comments:

Shrodinger —> Schrodinger throughout

paragraph line 146 is confusing. this equation is strange given that p(x^tar) and p(x^input) are dependent if taking batches.

**Limitations:**

I'm a bit confused about the limitations. It is claimed that

> One limitation of CLWF is that the samples obtained are not diverse enough as we use ODE for
> inference, which results in slightly higher test (continuous ranked probability score) CRPS compared
> to previous works, e.g., CSDI. Therefore, for future work, we will seek suitable approaches to
> accurately model the diffusion term in the SDE.

Other ODE models have had no problem with diversity. I would suggest that this is actually due to performing gradient ascent on the density of the VAE.

---

> ### Author Rebuttal · Authors · 2024-08-04
>
> Thank you very much for the constructive feedback and insightful comments
>
> ## Weaknesses (Quality) 1:
> We have conducted the ablation study on the impacts of sampling steps of CSDI and CLWF.
> The results show that CLWF has better performance when the simulation steps are small.
> Please also take a look at the uploaded PDF for the detailed experimental results.
>
> ## Weaknesses (Quality) 2:
> We have conducted additional experiment using the synthetic dataset.
> And the experimental results demonstrates the effectiveness of the Rao-Blackwellization, please see the uploaded PDF.
>
> ## Weaknesses (Quality) 3:
> We have added the error bars to the experimental results as suggested, please see the uploaded PDF.
>
>
> ## Weaknesses (Quality) 4:
> Compared to the SOTA methods, whose performances are already nearly optimal, the performance of our method has been significantly improved.
> In this situation, the Rao-Blackwellization can further enhance performance. Even without substantial improvements in values, it still demonstrates the effectiveness of our method.
>
> ## Weaknesses (Clarity):
>  ```
> As far as I can tell this “Rao-Blackwellized” sampler is fundamentally biased (while the original sampler is not), and it therefore cannot possibly be a Rao-Blackwellized sampler.
>  ```
> The Rao-Blackwell theorem only requires the base/original sampler is unbiased and the statistic is sufficient (not necessarily unbiased). According to the law of Total Expectation, the new sampler is also unbiased.
> Hence, one cannot say the new sampler is not Rao-Blackwellized.
> ## Justification for Rao-Blackwellization:
> Let's first consider a dynamic system which can be described by the following SDE:
> \begin{align}
> dX_t &= \mu_t(X_t,t)dt + \sigma_t dW_t,
> \end{align}
> where $\mu_t(X_t,t)$ is the velocity and can be solved via flow matching.
>
> Alternatively, if we assume the above system is a over-damped Langevin dynamic system, then system can be described by the following SDE:
> \begin{align}
> dX_t &= \nabla V_t(X_t,t)dt + \sigma_t dW_t
> \end{align}
> where $\nabla V_t(X_t,t)$ is the drift.
>
> Essentially, we have two different mathematical interpretations for the same system.
> Note that $\mu_t(X_t,t)$ (using flow matching) and $\nabla \mu_t(X_t,t)$ (using the potential function) are computed via different numerical methods, but ideally, they both suffice to represent the same system independently.
> We also assume the Gaussian potential function constructed via the VAE model is accurate enough.
>
> Since $X_{t+1}$ is the parameter we want to estimate, according to the Rao-Blackwell theorem, we construct the base sampler $\mathcal{S}$ based on the flow matching model $dX_t = \mu_t(X_t,t)dt + \sigma_t dW_t$ which is unbiased for $X_{t+1}$.
> And $\mathcal{T}(X_t)$ based on $dX_t = \nabla V_t(X_t,t)dt + \sigma_t dW_t$ is a sufficient statistic for $X_{t+1}$, meaning that $\mathcal{T}(X_t)$ contains all the necessary information for estimating $X_{t+1}$ but does not dependent on $X_{t+1}$.
> Afterwards, we define the new Rao-Blackwellized sampler $\mathcal{S}^* = \mathbb{E}[\mathcal{S}|\mathcal{T}]$ by letting the base sampler condition on the sufficient statistic.
>
>
> Now we prove that the new sampler is unbiased, i.e., $ \mathbb{E}[\mathcal{S}^*] = \mathbb{E}[\mathbb{E}[\mathcal{S}|\mathcal{T}]] = \mathbb{E}[\mathcal{S}]$, based on the Law of Total Expectation (or the Tower Rule).
> \begin{align}
> \mathbb{E}[\mathcal{S}^*] = \mathbb{E}[\mathbb{E}[\mathcal{S}|\mathcal{T}]] & = \iint s p(s|t)ds p(t)dt \nonumber \\
> & = \iint s p(s|t) p(t)dsdt = \iint s p(s,t) dsdt = \int s p(s) dt = \mathbb{E}[\mathcal{S}].
> \end{align}
> Since $\mathbb{E}[\mathcal{S}]$ is unbiased, $\mathbb{E}[\mathbb{E}[\mathcal{S}|\mathcal{T}]]$ is unbiased as well.
>
> By doing so, we can leverage the denoising/self-correction effect of the VAE model (considering that it is trained on clean data), which enables the reconstructed intermediate samples closer to the ground truth during the sampling process.
> If the above explanation is still not intuitive enough, one can think about a Kalman filter, whereby the first model is the system's dynamics and the second model is the noisy observation.
> The noise observation can be used to improve the trajectory estimation during tacking as long as it is sufficient.
> Please see new experimental results reported in the uploaded PDF.
> From the experiments results, we can see that the variances are reduced, and the improvement can be verified by calculating the RMSE.  For the future work, we will explore new techniques to further improve the stability of the Rao-Blackwellized sampler.
>
> ## Weaknesses (Significance):
> In this paper, we propose a novel framework based on Lagrangian dynamics and optimal transport for time series imputation and a novel sampler to enhance its performance. Our contributions are both theoretical and applied. We have also added new experimental results as the reviewer suggest, please refer to the uploaded PDF for more details.
>
> ## Questions 1:
>     ```
>         How is this a Rao-Blackwellization, and what does the paper
>     ```
> Since the comment does not fully display, we will only reply the first part of the question. Please see the above response to the issue regarding the Rao-Blackwellization.
>
> ## Questions 2:
> We have corrected the typos as suggested.
>
> ## Questions 3:
>  ```
> paragraph line 146 is confusing. this equation is strange given that $p(x^tar)$ and $p(x^input)$ are dependent if taking batches.
> ```
> The authors are confused by the comment as paragraph line 146 is not an equation.
> Doe the reviewer refers to Eq. (14)? This is the probabilistic coupling in context of optimal transport.
> Please refer to [1].
>
> ## Reference:
> [1] Xingchao Liu, Chengyue Gong, and Qiang Liu. Flow straight and fast: Learning to generate and transfer data with rectified flow. In The Eleventh International Conference on Learning Representations, 2022.

---

> ### Author Response · Authors · 2024-08-04
> **continuation of rebuttal**
>
> ## Limitations:
> '''
> Other ODE models have had no problem with diversity. I would suggest that this is actually due to performing gradient ascent on the density of the VAE.'''
> Other ODE models also have the problem with diversity to some extent.
> The usage of the VAE is not relevant on this issue, which, on the contrary, will slightly increase the randomnesses of the sampling procedure as the VAE is a probabilistic model.
> To clarify the statement, let's consider an ODE sampler described by
> \begin{align}
> X_T = X_0 + \int_0^T \mu (X_t,t)dt,
> \end{align}
> and an SDE sampler described by
> \begin{align}
> X_T = X_0 + \int_0^T \mu (X_t,t)dt  +  \int_0^T \sigma (X_t,t)dW_t.
> \end{align}
>
> From the above equations you can see that the randomness of the ODE sampler only arises from $X_0$ (the initial random noise) as the sampling process is deterministic.
> Thus the diversity of the samples obtained is limited as the velocity model $\mu (X_t,t)$ is biased towards the training data, which may suffer the overfitting issue.
> Please note that the training targets are only some finite samples of the target distribution but not the target distribution itself.
> On the contrary, the SDE-based sampling procedure can result in different samples even the initial value $X_0$ is the same datapoint due to the existence of the diffusion term $\sigma_t$.

---

> ### Author Response · Authors · 2024-08-04
> **continuation of rebuttal**
>
> ## Test imputation results on synthetic data ($5$ trials, values are multiplied by $10^2$.
>
>
> | Method  ||   Synthetic 0.4   ||  Synthetic 0.6  ||  Synthetic 0.8    |
> | ----------- |----------- | ----------- | ----------- | -----------   | -----------  | -----------  |
> |   | RMSE |  MAE  |RMSE |  MAE  |RMSE |  MAE  |
> | CLWF(no RB) |$22.91\pm0.49$ |$15.28\pm0.21$|$25.65\pm0.31$ |$15.54\pm0.22$|$27.41\pm0.27$  |$15.91\pm0.23$ |
> |CLWF(with RB)  | $22.72\pm0.48$| $13.23\pm0.42$|$25.44\pm0.30$|$15.28\pm0.17$| $27.32\pm0.27$| $15.79\pm0.23$ |
>
>
>
>
> ## Test imputation results on PM 2.5 with different simulation steps ($5$ trials)
>
>
> | Method  ||   5      steps   ||   10 steps   ||  15 steps    ||  20 steps     |
> | ----------- |----------- | ----------- | ----------- | -----------   | -----------  | -----------  | -----------  | -----------   |
> |   | RMSE |  MAE  |RMSE |  MAE  |RMSE |  MAE  |RMSE |  MAE  |
> | CSDI      |$34.21\pm0.16$  |$14.85\pm0.01$  |$29.43\pm0.46$   |$12.48\pm0.08$  |$22.40\pm0.16$ | $10.78\pm0.04$  |$19.22\pm0.13$  | $9.91\pm0.02$ |
> | CLWF      |$18.29\pm0.002$ |$9.78\pm0.004$  |$18.28\pm0.003$  |$9.77\pm0.005$  |$18.26\pm0.006$|$9.76\pm0.004$ |$18.21\pm0.002$ |$9.72\pm0.004$ |
>
>
>
>
>
> ## Test imputation results on PhysioNet 0.1 with different simulation steps ($5$ trials).
>
>
> | Method  ||   5 steps   ||   10 steps   ||  15 steps    ||  20 steps     |
> | ----------- |----------- | ----------- | ----------- | -----------   | -----------  | -----------  | -----------  | -----------   |
> |   | RMSE |  MAE  |RMSE |  MAE  |RMSE |  MAE  |RMSE |  MAE  |
> | CSDI      |$0.60\pm0.00$|$0.22\pm 0.000$|$0.58\pm0.005$|$0.22\pm0.000$|$0.57\pm0.002$ |$0.22\pm0.000$ |$0.56\pm 0.002$|$0.22\pm0.001$|
> | CLWF      |$0.48\pm0.000$|$0.22\pm0.00$|$0.47\pm0.000$|$0.22\pm0.00$|$0.47\pm0.000$|$0.22\pm0.00$  |$0.48\pm0.000$|$0.22\pm0.000$|
>
>
>
>
>
> ## Test imputation results on PhysioNet 0.5 with different simulation steps ($5$ trials).
>
>
> | Method  ||   5 steps   ||   10 steps   ||  15 steps    ||  20 steps     |
> | ----------- |----------- | ----------- | ----------- | -----------   | -----------  | -----------  | -----------  | -----------   |
> |   | RMSE |  MAE  |RMSE |  MAE  |RMSE |  MAE  |RMSE |  MAE  |
> | CSDI      |$0.71\pm0.000$  |$0.31\pm0.000$ |$0.69\pm0.001$  |$0.31\pm0.000$ |$0.68\pm0.001$ |$0.31\pm0.000$ |$0.68\pm0.000$  |$0.30\pm0.000$ |
> | CLWF      |$0.64\pm0.000$|$0.29\pm0.000$|$0.64\pm0.000$|$0.29\pm0.000$|$0.64\pm0.000$|$0.29\pm0.000$|$0.64\pm0.000$|$0.29\pm0.000$ |
>
>
>
>
> ## Test imputation results with error bars ($5$ trials).
>
>
> | Method  ||   PM 2.5   || PhysioNet 0.1 ||  PhysioNet 0.5    |
> | ----------- |----------- | ----------- | ----------- | -----------   | -----------  | -----------  |
> |   | RMSE |  MAE  |RMSE |  MAE  |RMSE |  MAE  |
> | CLWF(no RB) |$18.27\pm 0.01$ |$9.76\pm0.01$  |$0.47\pm 0.000$  |$0.22\pm 0.000$  |$ 0.64\pm0.002$  |$0.29\pm0.009$|
> |CLWF(with RB) | $18.08\pm 0.02$ |$9.71\pm 0.00$|

---

> > ### Comment · Reviewer_3hN3 · 2024-08-12
> >
> > I thank the authors for their detailed response, and additional experiments. Particularly when a small number of steps are used.
> >
> > > Justification of Rao-Blackwellization
> >
> > After the authors' response, I am fairly convinced this **not** a valid Rao-Blackwellization. The provided is not a valid proof that this is a valid Rao-Blackwellization, only that Rao-Blackwell samplers in general are valid.
> >
> > More specifically I believe at least two things are necessary to show that this is a valid Rao-Blackwellization:
> >
> > Why is $\mathbb{E}[\mathcal{S} | \mathcal{T}] = \mu_t^\theta(x^{input}, t) \frac{T}{N} + v_t^\phi(x_t, t) \frac{T}{N}$? This does not seem possible to me. Lets take the simple example of an OT-FM model with $p_0 = N(0,1)$ and $p_1 = N(1, 1)$ then the optimal $\mu_t^\theta(x, t)$ which we will call $\mu_t^\star(x) = 1$. $\nabla_x U(X_t) = \nabla \log N(t, 1) = t - x_t$. In fact any non-zero quantity will bias our estimate of $X_{t+1} | X_t = \mu_t^\star(x) = 1.$  Furthermore, following the suggested “Rao-Blackwellized” estimator, will not lead to $p_1$. It will be over concentrated towards the mean at every step.
> >
> > Why is $v_t^\psi(x_t, t) \frac{T}{N}$ a sufficient statistic? It seems impossible as the VAE does not depend on $t$ for it to be a sufficient statistic for $X_{t+1} | X_t$ which is time dependent.
> >
> > > Other ODE models also have the problem with diversity to some extent.
> > >
> >
> > This is simply not true. For every SDE there is an equivalent ODE with equivalent diversity through the probability flow ODE for any diversity metric measured on $p_T$.
> >
> > The experimental results are still extremely limited. The fact that this “Rao-Blackwellization” can only be tuned to work on one of the two datasets is not convincing. Also why are the provided results different than those in the paper?
> >
> > > Note that the PhysioNet dataset does not have enough non-zero data points to train a valid VAE model, therefore we only construct the Rao-Blackwellized sampler for the PM 2.5 dataset.
> >
> > If this is a true Rao-Blackwellization, then the estimate should be **no worse** than the simple estimator. As the authors state, it may not help but it should not hurt. This is further evidence to me that this theoretical result is invalid.
> >
> > My feeling is that that CLWF without (”Rao-Blackwellization”), which is equivalent to OT-CFM (Tong et al. 2023) / Minibatch-OT FM (Pooladian et al. 2023), should work quite well in this setting. However, the “Rao-Blackwellization” is not proven anywhere theoretically and the response does not justify it sufficiently. Furthermore the experimental results are only applied on a single dataset for a method that should be no worse than the baseline. I keep my current score of 2. However, am open to substantially raising it if the authors can show that this is a valid Rao-Blackwellization.

---

> > > ### Author Response · Authors · 2024-08-12
> > > **reply to rebuttal**
> > >
> > > ## Q: Why is $\mathbb{E}[\mathcal{S} | \mathcal{T}] = \mu_t^\theta(x^{input}, t) \frac{T}{N} + v_t^\phi(x_t, t) \frac{T}{N}$?
> > > ### A:    The proposed sampling step is not  $\mathbb{E}[\mathcal{S} | \mathcal{T}] = \mu_t^\theta(x^{input}, t) \frac{T}{N} + v_t^\phi(x_t, t) \frac{T}{N}$ (which the reviewer framed), but $\mathbb{E}[\mathcal{S} | \mathcal{T}] = \mu_t^\theta(x_t + v_t^\phi(x_t, t) \frac{T}{N} , t) \frac{T}{N} $, which is fundamentally different.
> > >
> > > ## Q: the optimal $\mu_t^\theta(x, t)$ which we will call $\mu_t^\star(x) = 1$
> > > ### A: Why the optimal velocity is $\mu_t^\star(x) = 1$ in your case ($p_0 = N(0,1)$ and $p_1 = N(1, 1)$)? You certainly cannot generate the target distribution via $x_{t+1} = x_t + h$ (not converging, more iterations always result in larger values).
> > >
> > >
> > >
> > > ## Q:  every SDE there is an equivalent ODE
> > > ### A: An ODE is always deterministic, which certainly limits its diversity to some extent.
> > >
> > >
> > > ## Q: It will be over concentrated towards the mean at every step.'''
> > > ### A: That is why this is an ODE sampler.
> > >
> > > ## Q: It seems impossible as the VAE does not depend on for it to be a sufficient statistic for which is time dependent.
> > > ### A: Please refer to the concepts of gradient flow and Langevin dynamics.
> > >
> > >
> > > ## Q: The experimental results are still extremely limited. The fact that this “Rao-Blackwellization” can only be tuned to work on one of the two datasets is not convincing. Also why are the provided results different than those in the paper?"
> > >
> > > ### A: We added additional experiments, please check the results carefully.
> > >
> > > ## Q: Also why are the provided results different than those in the paper?
> > > ### A: Because the learning process and sampling process are random like any other stochastic learning methods.

---

> > > > ### Comment · Reviewer_3hN3 · 2024-08-12
> > > >
> > > > Thank you for the quick response.
> > > >
> > > > > The proposed sampling step is not $\mathbb{E}[\mathcal{S} | \mathcal{T}] = \mu_t^\theta(x^{input}, t) \frac{T}{N} + v_t^\phi(x_t, t) \frac{T}{N}$
> > > > (which the reviewer framed), but, $\mathbb{E}[\mathcal{S} | \mathcal{T}] = \mu_t^\theta(x_t + v_t^\phi(x_t, t) \frac{T}{N} , t) \frac{T}{N}$, which is fundamentally different
> > > >
> > > > Okay great. I agree this is fundamentally different. This is closer, but still needs to be proven in my opinion. I also don't understand how this lines up with Algorithm 2 which clearly states $\mathbb{E}[\mathcal{S} | \mathcal{T}] = \mu_t^\theta(x^{input}, t) \frac{T}{N} + v_t^\phi(x_t, t) \frac{T}{N}$. How is this implemented practically?
> > > >
> > > > > Why the optimal velocity is $\mu_t^\star(x) = 1$...
> > > >
> > > > This is the optimal velocity for a flow model which is integrated from time zero to time one.
> > > >
> > > > Because $x_1 = x_0 + \int_{t=0}^1 \mu_t^\star(x) dt =  x_0 + \int_{t=0}^1 1 dt = x_0 + 1$
> > > >
> > > > > An ODE is always deterministic, which certainly limits its diversity to some extent.
> > > >
> > > > Yes, conditioned on an initial point $x_0$, an ODE has limited diversity, however conditioned on $p_0$, it does not.
> > > >
> > > > > Please refer to the concepts of gradient flow and Langevin dynamics.
> > > >
> > > > How does this prove that the VAE is a sufficient statistic for $X_{t+1}$? Or does this not need to be shown.
> > > >
> > > > > We added additional experiments, please check the results carefully.
> > > >
> > > > I have. There are no additional experiments on the "Rao-Blackwellization" that I can see.

---

> > > > > ### Author Response · Authors · 2024-08-13
> > > > > **reply**
> > > > >
> > > > > Thanks for the quick reply.
> > > > >
> > > > >       How is this implemented practically?
> > > > >
> > > > > At each step we first update the sample via $x_{t+1} = x_{t} +  v_t^\phi(x_t, t) \frac{T}{N}$, then obtain the new sample using the previous sample with $x_{t+1} = x_{t} +   \mu_t^\theta(x_t, t) \frac{T}{N}$ (you can take this as a correction step).
> > > > >
> > > > >
> > > > >     Yes, conditioned on an initial point $x_0$, an ODE has limited diversity, however conditioned on $p_0$, it does not.
> > > > >
> > > > > It still does, because in this case the diversity of the samples obtained only relies on the initial noise (which also appears in the training samples), which certainly causes that 1) it has lost some level of diversity during the sampling process compared SDE, 2) and the overfitting issue is more serious (especially when the variance of  $p_0$ is small).
> > > > >
> > > > >
> > > > >     How does this prove that the VAE is a sufficient statistic for $X_{t+1}$? Or does this not need to be shown.
> > > > >
> > > > > It suggests that the potential function does not need to be time-dependent function (as in our case). And that the derivative of the potential function can represent the sampling dynamics, thus it is a sufficient statistic for $X_{t+1}$.
> > > > >
> > > > >
> > > > >     There are no additional experiments on the "Rao-Blackwellization" that I can see.
> > > > > Please see the new results from the first Table (in the previous comment and the uploaded PDF) with CLWF(no RB) and CLWF(with RB).
> > > > >
> > > > > ## Previous comments
> > > > >     $\nabla_x U(X_t) = \nabla \log N(t, 1) = t - x_t$.
> > > > > It should be $\nabla_x U(X_t) = \nabla \log N(x_t|1) = 1 - x_t$.
> > > > >
> > > > >     any non-zero quantity will bias our estimate $X_{t+1} | X_t = \mu_t^\star(x) = 1.$
> > > > > in this case the velocity is optimal, then the derivative of the  potential function will be zero, which will not affect the sampling result.

---

### Official Review · Reviewer_FtzN · 2024-07-14

**Soundness:** 3
**Presentation:** 3
**Contribution:** 2
**Rating:** 5
**Confidence:** 3

**Summary:**

The paper presents Conditional Lagrangian Wasserstein Flow (CLWF), a new method for time series imputation. Using (entropic) optimal transport theory and Lagrangian mechanics, CLWF generates high-quality samples. Enhanced with a Rao-Blackwellized sampler, CLWF incorporates prior information through a variational autoencoder. Experiments on real-world datasets show that CLWF performs competitively.

**Strengths:**

1. The paper is well-written, clear, and easy-to-follow.
2. The use of a Rao-Blackwellized sampler and a variational autoencoder to integrate prior information enhances the model's performance, providing a more robust imputation process.

**Weaknesses:**

1. The conditional generation process is trained through the simulation-free training of the Schrödinger Bridge, which limits the novelty of the CLWF.
2. The paper mentions that the samples obtained using the ODE-based inference method may lack diversity, potentially leading to higher continuous ranked probability scores (CRPS) compared to some previous works.
3. While the method is tested on two real-world datasets, broader evaluation across more diverse and challenging datasets could strengthen the validation of the approach.

**Questions:**

I have no other questions.

**Limitations:**

Yes

---

> ### Author Rebuttal · Authors · 2024-08-04
>
> Thank you for your valuable feedback.
>
> ## Weakness 1:
> Indeed, CLWF is trained through the simulation-free training of the Schrödinger Bridge. However, in this work, based on the Lagrangian mechanics framework, we specifically proposed the Rao-Blackwellized sampler to further enhance the model's performance for time series imputation tasks.
>
>
> ## Weakness 2:
> The ODE sampler is adopted to optimize the performance of our method for the time series imputation tasks.
> However, an SDE sampler is also applicable within our proposed framework.
>
>
> Since our method is implemented via an ODE sampler CRPS may not be a very suitable evaluation metric.
> [1] and [2] suggest that CRPS could lead to an incorrect assessment of overall model performance due to the overlook of the model's performance on each dimension of the data.
>
>
> For the future work, we plan to explore more advanced methods to improve the diversity of the samples to have better empirical performance regarding CRPS .
>
>
> ## Weakness 3:
> The datasets used in our experiments, PM 2.5 and Physio, are both highly representative in the field of time series imputation.
> We have also included additional experimental results using synthetic data to further demonstrate the advantages of our methods.
> Please refer to the uploaded PDF for more detailed experimental information.
>
>
> ## References:
>
> [1] Koochali, Alireza, Peter Schichtel, Andreas Dengel, and Sheraz Ahmed. "Random noise vs. state-of-the-art probabilistic forecasting methods: A case study on CRPS-Sum discrimination ability." Applied Sciences 12, no. 10 (2022): 5104.
>
> [2] Biloš, Marin, Kashif Rasul, Anderson Schneider, Yuriy Nevmyvaka, and Stephan Günnemann. "Modeling temporal data as continuous functions with stochastic process diffusion." In International Conference on Machine Learning, pp. 2452-2470. PMLR, 2023.

---

> > ### Comment · Reviewer_FtzN · 2024-08-12
> >
> > The authors' reply addresses most of my issues. I appreciate the clarification made by the authors, and I have no other concerns.

---

### Author Rebuttal · Authors · 2024-08-04

We would like to thank all the reviewers for their insightful comments and try our best to address all the questions regarding this work.

To this end, we have included additional experimental results from an ablation study.
Specifically, we conducted new experiments using synthetic data to validate the effectiveness of the Rao-Blackwellization.
We also employed the real-world datasets to demonstrate that CLWF achieves better performance with fewer simulation steps.

Furthermore, all results are presented with $5$ trials, showing the corresponding means and standard deviations.
Please refer to the uploaded PDF file for detailed  experimental results.

---

### Decision · Program_Chairs · 2024-09-25

**Decision:**

Reject

**Comment:**

This paper deals with the problem of time series imputation. It introduces a method that uses tools from transport theory, Lagrangian mechanics, and optimal transport to generate samples.

This paper received mixed reviews (1x accept, 2x weak accept, 1x borderline accept, 1x strong reject). After the authors' rebuttal, some reviewers stated that their concerns had been addressed and maintained their score. Reviewer 3hN3, however, points to some remaining flaws of the paper. After the rebuttal period, only Reviewer 3hN3 participated in the follow-up discussion between reviewers and AC, and thus a consensus among the reviewers was not reached.

Because of the mixed opinions, I had a careful look at the paper myself. Unfortunately, I found it does not meet the acceptance bar and therefore I encourage the authors to improve their manuscript using the received feedback and submit to another conference.

Besides the issues found by Reviewer 3hN3, I noted that the paper contains basic mistakes in the mathematical equations. As an example, consider Eq. (7) (in the "Background" section), which is:
$$\mathcal{K}(x_t, \mu_t, t) = \mathbb{E}_{p(X_t)} \Bigg[ \int_0^T \int  \frac{1}{2} ||\mu_t(x_t, t)||^2 dx dt$$
(where the inner integral is over $\mathbb{R}^d$).
+ First, note that the square bracket isn't closed - I assume that's a simple typo and that the closing bracket should be after $dt$.
+ Second, note that the inner integral has $dx$ but there is no $x$ in the integrand - I assume that it should be $dx_t$ instead.
+ Third, if we integrate out $x_t$, then the expectation $\mathbb{E}_{p(X_t)}[\ldots]$ would not depend on $x_t$ at all, and therefore it can be safely ignored as everything inside is a constant - which should obviously not be the case.
+ Fourth, after we integrate out both $x_t$ and $t$, the result does not depend on either $x_t$ or $t$, whereas the LHS of the equation clearly depends on both - it is written as $\mathcal{K}(x_t, \mu_t, t)$.

As another example of wrong mathematics, Eq. (8) suffers from similar issues.

Besides these mistakes, I would like to note that the paper contains some unsupported claims. Two examples are:
+ In Section 3.5 ("Rao-Blackwellized sampler"), the text says: *"Note that the ODE sampler alone is good enough to generate high-quality samples for time series imputation"*. This leaves it unclear why switching from SDE to ODE is okay, as it's never discussed further.
+ In Section 3.4 ("Potential function"), the text says: *"In terms of practical implementation, we parameterize $\nabla_x U(X_t)$ via a Variational Autoencoder (VAE)"*. Once again, it remains unclear why that is an okay thing to do, since the (time-independent) VAE is fitted to the observed data only, and it might suffer from some biases that would propagate further. Reviewer 3hN3 also raised this point: *"More fundamentally I'm unclear why effectively taking steps towards data density of a VAE trained on a related (but fundamentally different) data distribution is even a good idea [...]"*.

Another point, raised by Reviewer 3hN3 (and which I agree with), is the lack of justification of the Rao-Blackwellized sampler. Section 3.5 does not clarify why this is a valid sampler, since the expectation $\mathbb{E}[\mathcal{S} | \mathcal{T}]$ is not correct. Unfortunately, despite this being one of the main contributions of the paper, this concern remained unaddressed after the rebuttal phase.

Even if the Rao-Blackwellized sampler were not theoretically valid, it could still be empirically useful; however, the experimental results do not support that conclusion (there is only a single result on a single dataset with marginally better results). It is also suspicious that, despite considering two datasets, the Rao-Blackwellized sampler is consctructed for one of them only, ignoring the second one.

To conclude, I think the paper would need a substantial changes before it is ready for acceptance at a conference like NeurIPS.